# A bi-diffusion based layer-wise sampling method for deep learning in large graphs

## Abstract

The Graph Convolutional Network (GCN) and its variants are powerful models for graph representation learning and have recently achieved great success on many graph-based applications. However, most of them target on shallow models (e.g. 2 layers) on relatively small graphs. Very recently, although many acceleration methods have been developed for GCNs training, it still remains a severe challenge how to scale GCN-like models to larger graphs and deeper layers due to the over-expansion of neighborhoods across layers. In this paper, to address the above challenge, we propose a novel layer-wise sampling strategy, which samples the nodes layer by layer conditionally based on the factors of the bi-directional diffusion between layers. In this way, we potentially restrict the time complexity linear to the number of layers, and construct a mini-batch of nodes with high local bi-directional influence (correlation). Further, we apply the self-attention mechanism to flexibly learn suitable weights for the sampled nodes, which allows the model to be able to incorporate both the first-order and higher-order proximities during a single layer propagation process without extra recursive propagation or skip connection. Extensive experiments on three large benchmark graphs demonstrate the effectiveness and efficiency of the proposed model.

## 1 Introduction

In recent years, extending deep learning approaches to graph domain has attracted increasing research attention. One of the successful attempts is the Graph Convolutional Network (GCN) (Kipf & Welling, 2016), which has become a crucial tool for representation learning of graph nodes. Given a graph, GCN applies the connectivity structure of the graph as the convolution filter (also known as the neighborhood aggregator (Hamilton et al., 2017)) to compute node representations layer by layer. At each layer, the representation of each node is obtained by mixing the features of its neighbors. Then, the final output representations can be used for various downstream applications. There have been many GCN-based attempts in the literature which have achieved the state-of-the-art performance on many graph-based applications, such as node classification (Kipf & Welling, 2016; Hamilton et al., 2017; Veličković et al., 2017), link prediction (Zhang & Chen, 2018), and recommender systems (Ying et al., 2018), etc.

Although numerous GCN-like models have been developed, most of them only focus on shallow models (e.g. 2 layers) on relatively small graphs. In practice, it still remains a severe challenge how to scale GCN-like models to larger graphs and deeper layers, due to the uncontrollable neighborhood expansion across layers. On the one hand, since the graph convolution (or say neighborhood aggregation) on each node needs to gather the features of its neighbors, and recursively the convolution computation on each of these neighbors also requires to gather the features of their own neighbors, it will incur the "neighbor-explosion" problem. When training a deep-layers model, the number of "neighbors" (and thus the training time) can grow exponentially with respect to the number of layers. Even with a usual mini-batch training, it will also involve a large amount of data for every batch and make the training computationally infeasible. On the other hand, the graph convolution is essentially a smoothing operation (Li et al., 2018; Xu et al., 2018a) by mixing the features of a node and its nearby neighbors, and thus makes the representations of nodes in the same cluster similar. However, the recursively expansion of neighborhoods layer by layer will quickly cover a large portion of the graph, leading many distant nodes from irrelevant clusters to be overly smoothed. Such

"over-smoothing" problem makes it hard for GCN-like models to maintain consistently resonable results in deep layers.

To mitigate the neighborhood expansion, some state-of-the-art methods (e.g., (Hamilton et al., 2017; Chen et al., 2018a; Ying et al., 2018)) use the stochastic mini-batch training and node-wise sampling optimization, which ensure that only a small fixed number of neighbors are selected by one node in each layer. While these methods significantly speed up the GCNs training on large-scale graphs, the time complexity still grows exponentially with the GCN depth even if the sampling number is small, and thus the overhead of these methods can be very large when the layers go deep. Further, Chen et al. (2018b) and Huang et al. (2018) propose to accelerate the training of GCNs by using the layer-wise sampling instead of node-wise sampling, which ensures a total fixed number of nodes are sampled together by each layer instead of each node, restricting the time complexity linear to the GCN depth. However, although efficiently solving the "neighbor explosion" problem, these existing layer-wise sampling algorithms still suffer from the "over-smoothing" issue in practice. Their constructed mini-batches potentially become too sparse to achieve high accuracy (see Section 3 for details).

In this paper, to address the above challenges, we first show that a desirable layer-wise sampler should simultaneously considers both the influence of parent nodes of upper layer on the candidate nodes of lower layer and the reverse influence of lower candidates on the upper parents. Motivated by that, we then propose a novel layer-wise sampling strategy, which samples the nodes layer by layer conditionally based on the factors of the bi-directional diffusion between layers. By sampling nodes with high bi-directional influence between neighbor layers, our layer-wise sampler tends to construct a closely-associated mini-batch, and thus elegantly and naturally tackles both the "neighbor explosion" and "over-smoothing" problem, as well as the sparsity issue suffered in current layerwise sampling algorithms. Furthermore, since the layer-wise sampler mixes different-hop neighborhoods in each single layer, we apply the self-attention mechanism as the aggregator to flexibly learn suitable weights for different-hop neighbors during the training, which allows the model to be able to incorporate both the first-order and higher-order proximities during a single layer propagation process without extra recursive propagation or skip connection. Finally, we conduct extensive experiments on three large benchmark graphs under the inductive, supervised learning setting. The experimental results demonstrate that our proposed model consistently and considerably outperforms the comparative state-of-the-art baselines and exhibit that our model can be efficiently and effectively scaled to very deep layers on very large graphs.

## 2   RELATED WORK

In the past few years, inspired by the convolutional neural network (CNN), which has revolutionized various machine learning tasks with grid-like data, how to extend the convolution operation on graph-structured data has attracted increasing research attention. Among the existing graph-based convolution network models, an important stream of work is built on spectral graph theory (Bruna et al., 2013). The pioneer spectral approach proposed by Bruna et al. (2013) first defines the parameterized convolution operation in Fourier domain, inspired by graph Fourier transform. Later, Henaff et al. (2015) apply efficient spectral filter to realize localized filtering operation, and Defferrard et al. (2016) further speed up the graph convolution computation by fast localized filters based on Chebyshev expansion of the graph Laplacian. Recently, GCN (Kipf & Welling, 2016) is proposed to simplify the aforementioned spectral methods with first-order expansion and re-parameterization trick, which has achieved state-of-the-art performance on supervised/semi-supervised node classification task and has motivated many GCN-based variants and applications. For example, Veličković et al. (2017) replace the convolution operation by applying the attention mechanism (Vaswani et al., 2017) as the aggregator to capture neighbor features with adjustable trainable weights. Klicpera et al. (2018) combine PageRank with GCN to enable efficient information propagation from multiple hops away. Xu et al. (2018b) borrow the idea of "skip-connection" (He et al., 2016) into the GCN context to improve the accuracy of GCNs with more than two layers.

Although numerous GCN variants have been developed, most of them conduct the training in full batch, which limits the application of these methods only to small graphs. In order to scale GCNs to large graphs, the sampling-based algorithms have been recently proposed for efficient mini-batch training. For example, the node-wise sampling based algorithm GraphSAGE proposed by Hamilton

et al. (2017) computes node representations by randomly sampling neighborhoods of each node and performing a specific aggregator for information smoothing, in which the neighborhood aggregation operation can be regarded as an another perspective of graph convolution. Ying et al. (2018) enhance the sampler of GraphSAGE by introducing an importance score to each neighbor node, and thus lead to less information loss due to the preference towards influential neighbors. Although these methods ensure that only a small fixed number of neighbors are selected for each node in the next layer, they still require the recursive expansion of neighborhoods across layers, which leads to the total number of support nodes (and thus the training time) exponential with the GCN depth. To further restrict the neighborhood size of deep layers, Chen et al. (2018a) propose to use the historical activations in the previous layer to avoid redundant re-evaluation and thus reduce to only two neighbors to support the computation of next layer activations. However, it requires storing all the intermediate embeddings of all the nodes in memory, leading hefty memory requirement. Chen et al. (2018b) interpret graph convolutions as integral transforms of embedding functions and sample the nodes in each layer independently. They apply importance sampling for explicit variance reduction, and remarkably, their method leads to constant number of samples in all layers. Then, Huang et al. (2018) follow the idea of layer-wise sampling proposed in (Chen et al., 2018b) to extend to use an additional sampling neural network to sample nodes for the lower layer conditionally based on the nodes of upper layer instead of sampling independently. However, these layerwise sampling based methods potentially build up very sparse mini-batchs, especially when the GCN layers grow deeply. In addition, Zeng et al. (2019a) present a subgraph based training algorithm that is scalable with respect to GCN depth, and Chiang et al. (2019) propose to build mini-batches based on (topological) clusters of the training graph. These subgraph based methods, although empirically demonstrate the effectiveness on deep models, essentially tackle the scalability issue by dividing the original large graph into a series of small graphs, which inherently results in biased estimation of the full batch loss. In summary, with numerous GCN variants being developed, it still remains a question how to train these models (with very deep layers) efficiently on very large graphs.

## 3 NOTATIONS AND PRELIMINARIES

In this section, we first introduce some notations used throughout this paper and then explain the problem our model solves.

This paper mainly focuses on undirected attributed graphs. Let $\mathcal{G} = (\mathcal{V}, \mathcal{E})$ denote the undirected graph with nodes $v_i \in \mathcal{V}$, edges $(v_i, v_j) \in \mathcal{E}$, and $n = |\mathcal{V}|$ defines the number of the nodes, $m = |\mathcal{E}|$ defines the number of the edges. Let $A \in \mathbb{R}^{n \times n}$ denote the adjacency matrix of the graph with each entry $A_{i,j}$ equaling to 1 if there is an edge between $v_i$ and $v_j$ and 0 otherwise. $\tilde{A} = A + I_n$ denotes the adjacency matrix with added self-loops. Also, the graph $\mathcal{G}$ is attributed by a feature/attribute matrix $X \in \mathbb{R}^{n \times d}$ with $X_i$ denoting the $d$-dimensional feature/attribute vector for node $v_i$.

Given the undirected attributed graph, the problem we consider in this paper is representation learning for graph nodes by aggregating (or propagating) their features/attributes. The closest work to this vein is the Graph Convolution Network (GCN) (Kipf & Welling, 2016) and its variants. A GCN is a kind of multi-layer convolutional network, and it uses the connectivity structure of the graph as the convolution filter to perform neighborhood aggregation/propagation layer by layer. At each layer, the hidden representation of a node is obtained by aggregating the last hidden representations of its neighbors (and itself), followed by one or a few layers of linear transformations and nonlinear activations. The final output representations can then be used for downstream tasks. For example, in node classification task, the final output representations are fed to a classifier to predict node labels, and thus the parameters of GCN can be trained in an end-to-end manner. Let $h_v^{(l)} \in \mathbb{R}^{1 \times d^{(l)}}$ denote the hidden representation (with dimensionality $d^{(l)}$) of node $v$ in the $l$-th layer, the feed forward propagation rule of single layer in GCN is defined as follows:

$$h_v^{(l)} = \sigma \Big( \sum_{u \in N(v)} \hat{\tilde{A}}_{v,u} h_u^{(l-1)} W^{(l)} \Big), \qquad v \in \mathcal{V}, \, l = 1, 2, \cdots, L, \tag{1}$$

where $N(v)$ is the set of neighbors of node $v$, and note that we always consider the node itself as its self-loop neighbor; $\hat{\tilde{A}} = \tilde{D}^{-\frac{1}{2}} \tilde{A} \tilde{D}^{-\frac{1}{2}}$ is the symmetrically normalized adjacency matrix with self-loops, with the diagonal degree matrix $\tilde{D}_{i,i} = \sum_j \tilde{A}_{i,j}$; $W^{(l)} \in \mathbb{R}^{d^{(l-1)} \times d^{(l)}}$ is the trainable weight matrix in the $l$-th layer; $\sigma(\cdot)$ is the activation function (e.g. ReLU). Note that we use the

graph attributes as the initial representations, i.e. $h_v^{(0)} = X_v$. As a result, both the input graph attributes and structure are "embedded" into the final output representations in the $L$-th layer.

Since the GCN needs to recursively perform neighborhood aggregation across layers, there is an obvious challenge for applying current GCN model with deep layers—over-expansion of neighborhoods. In practice, the "over-expansion" issue incurs two severe problems in GCN training: "neighbor explosion" and "over-smoothing". First, considering the width of the neighborhood expansion layer by layer, it will incur the neighbor explosion problem. As the representation of a node at layer $l$ is computed recursively by aggregating the representations of its neighbors at layer $l - 1$, it is easy to see that the more layers we applied, the more multi-hop neighbors we need to support the computation of the root node. The number of support nodes can grow exponentially with respect to the number of layers. Particularly for dense graphs and powerlaw graphs, the expansion of the neighborhoods for a single root node will quickly involve a large number of support nodes, and thus incur expensive computations and memory footprints. Second, considering the depth of the over-expansion, it will lead to the over-smoothing problem. As highlighted in (Li et al., 2018) and (Xu et al., 2018a), the graph convolution (neighborhood aggregation) in GCN is simply a special form of Laplacian smoothing, which mixes the features of a node and its nearby neighbors. The smoothing operation makes the features of nodes in the same cluster similar, thus greatly easing the classification task. However, adding more layers to a GCN will make more distant "neighbors" to be mixed, which actually are irrelevant nodes. Even for a single node, it will quickly cover a large portion of the graph due to the neighborhood expansion layer by layer. As a result, the output representations with too many layers in GCN may be over-smoothed and nodes from different clusters may become indistinguishable.

To mitigate the above "over-expansion" issue, many sampling-based algorithms are very recently proposed to control the expansion of the neighborhoods. Hamilton et al. (2017) firstly attempt to use a nodewise sampling strategy to approximate the GCN model. Instead of the full expansion of neighborhoods for the feed forward computation of each node, the nodewise sampling strategy uniformly samples a small number of neighbors for each node at each layer. Then, the propagation rule of Eq. (1) is approximated as:

$$h_v^{(l)} = \sigma(\frac{|N(v)|}{|N_s(v)|} \sum_{u \in N_s(v)} \hat{\tilde{A}}_{v,u} h_u^{(l-1)} W^{(l)}), \quad v \in \mathcal{V}, \; l = 1, 2, \cdots, L, \quad (2)$$

where each element $u \in N_s(v)$ is uniformly sampled from the whole neighbor set $N(v)$. Although clearly reduces the receptive field size of graph convolution and achieves competitive performance, such nodewise sampling is still computationally expensive for deep networks, because the number of sampled neighbors still grows exponentially with the number of layers. Assuming the sample size for all layers is fixed as $s$, the number of sampling neighbors in the $l$-th layer will increase to $O(s^l)$, Even with a very small $s$, this will also lead to significant computational burden for large $l$.

FastGCN (Chen et al., 2018b) and AS-GCN (Huang et al., 2018) are another state-of-the-art sampling-based algorithms, which use a layerwise sampling strategy to avoid the neighborhood expansion. As opposed to the nodewise method in Eq. (2) which samples a fixed-size set of neighbors $N_s(v)$ for each parent node $v$ of upper layer independently, the layerwise sampling directly samples a fixed number of nodes for each layer altogether, i.e., the sampling for all the parent nodes of upper layer is jointly performed only once, and the sampled nodes are shared by all upper parent nodes. Formally, let $\mathcal{V}^{(l-1)}$ denote the set of nodes of $(l-1)$-th layer, which are generated by layerwise sampling to support the computations of nodes $\mathcal{V}^{(l)}$ of $l$-th layer, and the top layer nodes $\mathcal{V}^{(L)}$ are the pre-selected nodes (e.g. the stochastic mini-batch of nodes) whose final representations are used in downstream tasks. The propagation rule of GCN based on layerwise samlping is defined as:

$$h_v^{(l)} = \sigma(\frac{|N(v)|}{|\mathcal{V}^{(l-1)}|} \sum_{u \in \mathcal{V}^{(l-1)}} \hat{\tilde{A}}_{v,u} h_u^{(l-1)} W^{(l)}), \quad \mathcal{V}^{(l-1)} \sim q(u|\mathcal{V}^{(l)}), \; l = 1, 2, \cdots, L, \quad (3)$$

where $q(u|\mathcal{V}^{(l)})$ is defined as the probability of sampling $u$ given all the parent nodes of upper layer (i.e. $\mathcal{V}^{(l)}$), and $\mathcal{V}^{(l-1)}$ are all the nodes of lower layer sampled according to $q(u|\mathcal{V}^{(l)})$. In FastGCN, $q(u|\mathcal{V}^{(l)})$ is simply designed as: $q(u|\mathcal{V}^{(l)}) \propto \frac{1}{|N(u)|} \sum_{\hat{u} \in N(u)} \frac{1}{|N(\hat{u})|}$. In AS-GCN, $q(u|\mathcal{V}^{(l)}) = \frac{\sum_{v \in \mathcal{V}^{(l)}} \mathbf{1}_{u \in N(v)} |g(X_u)|}{\sum_{v \in \mathcal{V}^{(l)}} |N(v)| |g(X_v)|}$, where $g(X_u)$ is a linear function to compute the self-dependent factor based

on the node feature $X_u$. Simply, FastGCN contructs each layer independently according to an identical distribution, AS-GCN builds up the network layer by layer in a top-down way, where the nodes in the lower layer are sampled conditionally based on the upper layer's.

Based on the layerwise sampling, it is easy to fix the size of each layer to avoid the "neighbor explosion", as the total number of nodes only grows linearly with the network depth. Nonetheless, the current layerwise sampling strategies used in FastGCN and AS-GCN still confront the "over-smoothing" problem and potentially build up very sparse mini-batchs. Formally, FastGCN and AS-GCN only sparsely subsample the potential "neighbors" during the expansion of the neighborhood across layers. As a result, although the neighbor explosion problem is easily solved by potentially restricting the factor of the width of expansion to 1, it can also quickly reach the very deep part of the graph due to the "over expansion" of the depth layer by layer, and thus overly smooth the irrelevant nodes from distant clusters. Moreover, by only subsampling the potential neighbors in each layer without controlling the depth of "over-expansion", the above layerwise sampling strategies will contruct very sparse minibatchs in practice. As the correlations among nodes will decrease rapidly with the increasing of the depth, in very deep layers, the between-layer connectivity will be very sparse and many parent nodes of upper layer may have no connected neighbors, which makes the GCN difficult to achieve high accuracy.

In the rest of this paper, we present a desirable layerwise sampling algorithm, elegantly and naturally tackling the above problems of existing layerwise samping.

## 4 PROPOSED METHOD: BLS-GAN

To mitigate the over-expansion issue in deep graph neural networks, in this section, we present a novel layerwise sampling strategy, which samples the nodes layer by layer conditionally based on the factors from the bi-directional diffusion processes between layers. Next, we leverage the self-attention mechanism to flexibly aggregate the sampled nodes in each layer with adaptive learnable weights, and finally propose our Bi-diffusion guided Layer-wise Sampling based Graph Attention Network (called BLS-GAN) model.

### 4.1 BI-DIFFUSION BASED LAYERWISE SAMPLING

As analyzed in Section 3, a desirable sampling algorithm should not only control the width of neighborhood expansion across layers, i.e., the size of sampled nodes in each layer to avoid the "neighbor explosion", but also restrict the depth of neighborhood expansion, i.e., the distance among nodes of inter-layers to avoid the "over-smoothing" and the "sparse-connectivity". The idea of layerwise sampling naturally meets the first requirement, as the nodes of the lower layer are sampled as a whole. However, the current layerwise sampling strategies either contruct each layer independently according to an identical distribution (Chen et al., 2018b), or one-sidedly sample nodes in the lower layer proportionally to the connections to the upper layer's (Huang et al., 2018). Such layer samplers could not satisfy the second requirement and potentially return overly sparse minibatchs when the network is deep (Zeng et al., 2019b). To address the above problems, in this paper, we present a novel desirable layerwise sampling strategy, which satisfies both the two requirements.

To control the depth of neighborhood expansion in layerwise sampling, it is natural and reasonable to sample nodes with high local correlation. However, we highlight that, the connected edges, which used in current layerwise sampling algorithms, do not proportionally reflect the real correlation among nodes. When sampling nodes in the lower layer conditioned on the upper layer's, the current start-of-the-art layer sampler usually uses higher probabilities to sample nodes who have more connected edges to the parent nodes of upper layer. However, the sampled nodes with many connections can have weak correlation. For example, in biological and citation networks, the majority of the nodes have few connections, whereas some core nodes and hub nodes are usually connected to many other nodes. That is, such cores or hubs, although have many connected edges to the parent nodes of upper layer, do not have a strong correlation with the upper layer's, because there is few "influence" of the upper parent nodes on those cores or hubs. In practice, due to the weak influence (i.e. correlation), such cores or hubs are not worthwhile candidates. Specially, if sampling them, in next layers, the sampler will quickly expand onto a very large portion of the graph, where many nodes are irrelevant.

The bias between the correlation and connections is due to the fact that correlation (or say influence) is bi-directional. The aforementioned connections to the parent nodes of upper layer can only indicate the influence from the sampling nodes of lower layer to the upper layer's, but not reflect the reverse influence. Therefore, a desirable layer sampler should select nodes in the lower layer conditionally based on the bi-directional influences on the parent nodes of upper layer. Below we describe a principled way to achieve it.

We define the "influence" (also corelation) from the graph connectivity perspective and estimate it by developing the diffusion process (i.e. the random walk simulation) on the graph. Formally, Let $P \in \mathbb{R}^{n \times n}$ denote the diffusion matrix (i.e. the transition probability matrix of random walk) of the graph, we have $P = D^{-1}A$ where $A$ is the adjacency matrix of the graph, $D$ is the diagonal degree matrix with $D_{i,i} = \sum_j A_{i,j}$. Then, given all the parent nodes in the upper layer (denoted as $\mathcal{V}^l$), the diffusion factor from an arbitrary node $u$ in the lower layer to the upper nodes $\mathcal{V}^l$ is defined as:

$$\gamma(u, \mathcal{V}^{(l)}) = \frac{\sum_{\hat{v} \in \mathcal{V}^{(l)}} P_{u,\hat{v}}}{\sum_{v \in \mathcal{V}} P_{u,v}} = \sum_{\hat{v} \in \mathcal{V}^{(l)}} P_{u,\hat{v}}. \tag{4}$$

Inversely, the diffusion factor from $\mathcal{V}^l$ to the candidate node $u$ is defined as:

$$\lambda(\mathcal{V}^{(l)}, u) = \frac{\sum_{\hat{v} \in \mathcal{V}^{(l)}} P_{\hat{v},u}}{\sum_{\hat{v} \in \mathcal{V}^{(l)}} \sum_{v \in \mathcal{V}} P_{\hat{v},v}} = \frac{\sum_{\hat{v} \in \mathcal{V}^{(l)}} P_{\hat{v},u}}{|\mathcal{V}^{(l)}|}. \tag{5}$$

Intuitively, the diffusion factor from the candidate node of lower layer to the parent nodes of upper layer indicates the influence from the upper layer nodes to the candidate node. The larger the factor, the higher the influence of the upper layer's on the candidate. Reversely, the diffusion factor from the parent nodes of upper layer to a candidate node reflects the influence from the candidate to the upper layer's, and the larger the factor, the higher the influence of the candidate on the upper layer's. Clearly, a desirable layerwise sampler should incorporate the factors in the above bi-directional diffusion processes to sample nodes with high bi-directional influence on each other. Therefore, we define the probability distribution of sampling a node $u$ in the lower layer given all the nodes $\mathcal{V}^{(l)}$ of the upper layer as:

$$P(u|\mathcal{V}^{(l)}) = \frac{\gamma(u, \mathcal{V}^{(l)}) \cdot \lambda(\mathcal{V}^{(l)}, u)}{\sum_{\hat{u} \in \mathcal{V}} \gamma(\hat{u}, \mathcal{V}^{(l)}) \cdot \lambda(\mathcal{V}^{(l)}, \hat{u})}. \tag{6}$$

Then, according to this probability distribution, we can easily sample an arbitrary number of nodes to build up the lower layer, and the feed forward propagation between layers can be consistently computed as the Eq. (3) by updating the $q(u|\mathcal{V}^{(l)})$ in Eq. (3) as the $P(u|\mathcal{V}^{(l)})$ of Eq. (6).

We term the above sampling as the bi-diffusion (bi-directional diffusion) based layerwise sampling strategy. By sampling the nodes layer by layer according to the bi-diffusion factors between layers, the sampler tends to construct a mini-batch of nodes with high local bi-directional influence/correlation, and thus naturally tackle the "over-expansion" issue across layers, in view of both the width and the depth of neighborhood expansion. The former view can speed up training of a deep-layers model by avoiding the "neighbor-explosion"; The latter view potentially enhances the model's performance by mitigating the "over-smoothing" and the "sparse-connectivity".

## 4.2 SELF-ATTENTION AGGREGATOR

Although the idea of training by the layerwise sampling is applicable to many GCN variants, in this section, we leverage the self-attention mechanism to flexibly aggregate the sampled nodes in each layer with adaptive learnable weights, which allows the model to be able to simultaneously exploit both the first-order and higher-order proximities during a single layer propagation process.

By applying the layer-wise sampling layer by layer, the nodes of lower layer are sampled together and the sampled nodes are shared by all parent nodes of upper layer. That is, the sampled nodes are not always the 1-hop neighbors (with direct connnections) of the upper parent nodes. Therefore, the bi-diffusion based layerwise sampling proposed in Section 4.1, is essentially a bi-diffusion based mix-hop sampling approach, which samples the highly relevant nodes from both 1-hop and multi-hop neighborhoods by jointly treating the upper parent nodes as a whole. As a result, the higher-order proximity is maintained in each single layer without extra recursive propagation or skip connection. When we perform bottom-up propagations to update the hidden features for the nodes of

upper layers (like Eq. (3)), the updater for each node in each layer always simultaneously aggregates messages passed from both 1-hop neighborhood and multi-hop neighborhood, thus enabling more efficient layer propagation and model training. Further, considering that the proximities of different orders measure the corelation of nodes from different levels of scope, the neighbors of different hops should not be treated equivalently during the layer aggregation. However, the aggregation weight in GCN aggregator (i.e. the term $\hat{\tilde{A}}_{v,u}$ in Eq. (3)) is strictly designed as some normalization of the graph adjacency matrix, which can be only suitable for expressing the first-order proximity. Intuitively, the aggregation weights for $k$-hop neighbors are supposed to be some normalization of the $k$-th power of the adjacency matrix, which is computationally infeasible. Therefore, to avoid the expensive computation of mixing powers of the adjacency matrix, we follow (Veličković et al., 2017) to apply the self-attention mechanism as the aggregator, which is able to flexibly learn suitable weights for different-hop nodes during the training. Concisely, we replace the aggregation weight $\hat{\tilde{A}}_{v,u}$ in Eq. (3) with specific self-attention scores, and update the propagation rule as:

$$h_v^{(l)} = \sigma\big( \sum_{u \in \mathcal{V}^{(l-1)} \cup \{v\}} \alpha(h_v^{(l-1)}, h_u^{(l-1)}) h_u^{(l-1)} W^{(l)} \big), \quad \mathcal{V}^{(l-1)} \sim P(u|\mathcal{V}^{(l)}), \, l = 1, 2, \cdots, L, \quad (7)$$

where $\alpha(h_v^{(l-1)}, h_u^{(l-1)})$ measures the self-attention scores between the parent node $v$ and the layer-wise sampled node $u$ according to their hidden features in layer $l-1$, and is defined as:

$$\alpha(h_v^{(l-1)}, h_u^{(l-1)}) = \frac{\exp(\text{LeakyReLU}([h_v^{(l-1)}W \parallel h_u^{(l-1)}W]\Theta))}{\sum_{u \in \mathcal{V}^{(l-1)} \cup \{v\}} \exp(\text{LeakyReLU}([h_v^{(l-1)}W \parallel h_u^{(l-1)}W]\Theta))}, \quad (8)$$

where $\parallel$ is the concatenation operation, $W \in \mathbb{R}^{d^{(l-1)} \times d^{(l-1)}}$ and $\Theta \in \mathbb{R}^{2d^{(l-1)} \times 1}$ are trainable parameters.

Overall, given a mini-batch of nodes as the top layer and the number of layers $L$, we can apply the bi-diffusion based layerwise sampling (Eq. (6)) to recursively construct $L$ layers of support nodes in a top-down way, and then stack $L$ self-attention aggregators (Eq. (7)) between each two layers to perform bottom-up propagations. At each layer, the hidden representation of a node is updated by flexibly aggregating the features of its multi-hop neighbors (and itself) with trainable weights. The final output representations in the top layer are then used for downstream applications. As a result, we obtain a novel deep graph neural network model, which could efficiently and effectively embed both the graph attributes and structure into the output representations. We call the proposed model as BLS-GAN (Bi-diffusion guided Layer-wise Sampling based Graph Attention Network). Finally, as an example in practice, we apply the BLS-GAN for node classification in this paper, which is the most popular application of graph neural network. In this setting, the final output representations of nodes are fed to a classifier to predict node labels, and the parameters of BLS-GAN are trained in an end-to-end manner by using iterative algorithms (e.g. mini-batch SGD) to minimize the following loss function:

$$\mathcal{L} = \frac{1}{|\mathcal{V}|} \sum_{v \in \mathcal{V}} \mathcal{L}_c(y_v, f(h_v^{(L)})), \quad (9)$$

where $h_v^{(L)}$ is the final output representation of node $v$ learned by BLS-GAN, $f(\cdot)$ is a simple single-layer feed-forward neural network classifier, $\mathcal{L}_c$ is the classification loss (e.g., the crossing entropy) for node $v$ to predict its ground-truth label $y_v$.

## 5 EXPERIMENTS

In this section, we report experimental results to demonstrate the effectiveness and efficiency of our proposed BLS-GAN model. The code will be released in the camera-ready version due to the double-blind review policy.

### 5.1 EXPERIMENTAL SETUP

We follow the experiment setup in GraphSAGE (Hamilton et al., 2017) and FastGCN (Chen et al., 2018b) to compare with the state-of-the-art GCN training algorithms under the inductive, supervised learning setting. We use three large graph datasets: Reddit, PPI, and Pubmed, which are publicly

Table 1: Statistics of the datasets used in our experiments.

| Dataset | Nodes | Edges | Classes | Features | Training/Validation/Test |
|---------|-------|-------|---------|----------|--------------------------|
| Reddit  | 232,965 | 11,606,919 | 41 | 602 | 152,410/23,699/55,334 |
| PPI     | 56,944  | 818,716    | 50 | 121 | 44,906/6,514/5,524 |
| Pubmed  | 19,717  | 44,338     | 3  | 500 | 18,217/500/1,000 |

available in the aforementioned references. In the Reddit graph, a node represents a post on Reddit, and an edge is formed if the posts share a common user in their comments. The node features (attributes) are the concatenation of word vectors of their title and comments. The classes are the community of the posts. In the PPI (protein-protein interaction) graph, the features correspond to positional gene sets, motif gene sets and immunological signatures (Zitnik & Leskovec, 2017). The classes are gene ontology sets. In the Pubmed dataset, nodes are documents, edges are citation links, and classes are research topics of documents. Node features correspond to elements of a bag-of-words representation of documents. Under the inductive setting, all three datasets are splited into three partitions (training/validation/test), and such a split is coherent with that used in GraphSAGE and FastGCN. Critically, all the testing nodes (including their edges and features) remain completely unobserved during training. Under the supervised learning (node classification) setting, all labels of the training nodes are used for training. The statistic of the datasets are summarized in Table 1.

We compare the proposed BLS-GAN model with five state-of-the-art baselines, including the vanilla GCN (Kipf & Welling, 2016) and GAT (Veličković et al., 2017) models (with their mini-batched implementations), the nodewise sampling based method: GraphSAGE (Hamilton et al., 2017), and the layer-wise sampling based algorithms: FastGCN (Chen et al., 2018b) and AS-GCN (Huang et al., 2018). As introduced in Section 3, GraphSAGE randomly samples fixed-size neighbors for each node at each layer, FastGCN contructs each layer independently according to an identical distribution, AS-GCN samples nodes of lower layer conditionally based on the upper layer's. Note that to make the comparisons more fair, we re-implement all the baselines based on our framework and all the codes will be released after reviw.

For all datasets, we train both our BLS-GAN and the baselines by minimizing the cross-entropy on the training nodes using the mini-batch Adam SGD optimizer with the batch size of 256 and the initial learning rate of 0.005. We set the maximum of training epochs as 200 and use an early stopping strategy with a window size of 10 on the Micro-F1 scores on the validation nodes. After training, we compare BLS-GAN with baselines by evaluating the Micro-F1 scores of classification/prediction on the test nodes. For all models, we implement them with adjustable depth (layers) of [2,3,4,5] to detailly demonstrate their performance by stacking different layers. The hidden dimensions in all layers are fixed as 256. We also equalize the sample size on all layers: the nodewise sample size for each node in GraphSAGE is fixed as 10, the layerwise sample size for each layer in our BLS-GAN and FastGCN and AS-GCN is fixed as 100. Specially, we follow GAT to employ multi-head attention to stabilize the learning process of self-attention, and the head size are set as 4.

## 5.2 RESULTS

We compare our proposed BLS-GAN model with the state-of-the-art baselines by evaluating both the effectiveness (classification Micro-F1 score) and efficiency (training time) on all the three benchmark datasets. The results are shown in Table 2 and Table 3.

Table 2 shows the Micro-F1 scores for node classification using different models with different layers. We can observe that the proposed BLS-GAN consistently outperforms all the start-of-the-art baselines on all datasets, and particularly in deep layers (e.g. 5), BLS-GAN achieves significant gains compared to other approaches. In most cases, the deeper the layers, the greater the gains. For example, on the Reddit dataset, BLS-GAN achieves the improvements by 0.01–0.04 (relatively 1%–5%) over all the five start-of-the-art baselines in the setting of 2 layers, and by 0.03–0.08 (relatively 3%–10%) over all baselines in the setting of 5 layers. Consistently, on the PPI dataset, among different layers, we can find that BLS-GAN improves the Micro-F1 scores by 0.20–0.25 (relatively 26%–35%) over GCN, by 0.005–0.08 (relatively 0.5%–9%) over GAT, by 0.20–0.30 (relatively 26%–45%) over GraphSAGE, by 0.30–0.46 (relatively 47%–87%) over FastGCN, and by 0.21–0.24 (relatively 28%–33%) over AS-GCN. Specially, it is easy to see that by stacking more

Table 2: The Micro-F1 scores for node classification using different models with different layers.

| Dataset | #Layers | GCN | GAT | GraphSAGE | FastGCN | AS-GCN | **BLS-GAN** |
|---|---|---|---|---|---|---|---|
| Reddit | 2 | 0.9381 | 0.9356 | 0.9297 | 0.9109 | 0.9289 | 0.9507 |
|  | 3 | 0.9468 | 0.9485 | 0.9386 | 0.9023 | 0.9307 | 0.9542 |
|  | 4 | 0.9335 | 0.9423 | 0.9312 | 0.8809 | 0.9247 | 0.9564 |
|  | 5 | 0.9020 | 0.9264 | 0.9185 | 0.8732 | 0.9138 | 0.9569 |
| PPI | 2 | 0.7130 | 0.8851 | 0.6671 | 0.6539 | 0.7325 | 0.9637 |
|  | 3 | 0.7541 | 0.9547 | 0.7191 | 0.6260 | 0.7619 | 0.9755 |
|  | 4 | 0.7780 | 0.9779 | 0.7789 | 0.5860 | 0.7675 | 0.9835 |
|  | 5 | 0.7407 | 0.9461 | 0.7520 | 0.5256 | 0.7408 | 0.9850 |
| Pubmed | 2 | 0.8788 | 0.8786 | 0.8765 | 0.8582 | 0.8724 | 0.8856 |
|  | 3 | 0.8854 | 0.8863 | 0.8843 | 0.8512 | 0.8815 | 0.8923 |
|  | 4 | 0.8714 | 0.8894 | 0.8746 | 0.8491 | 0.8755 | 0.8977 |
|  | 5 | 0.8546 | 0.8769 | 0.8718 | 0.8173 | 0.8621 | 0.8981 |

Table 3: The speedups compared with GCN on the three benchmark datasets.

| Dataset | #Layers | GCN | GAT | GraphSAGE | FastGCN | AS-GCN | **BLS-GAN** |
|---|---|---|---|---|---|---|---|
| Reddit | 2 | 1.00 | 0.89 | 4.32 | 3.50 | 3.79 | 3.63 |
|  | 3 | 1.00 | 0.90 | 3.16 | 11.08 | 11.67 | 10.26 |
|  | 4 | 1.00 | 0.93 | 2.98 | 34.56 | 30.48 | 28.75 |
|  | 5 | 1.00 | 0.92 | 1.53 | 66.88 | 53.36 | 48.91 |
| PPI | 2 | 1.00 | 0.94 | 1.00 | 0.81 | 0.74 | 0.71 |
|  | 3 | 1.00 | 0.97 | 0.76 | 1.98 | 1.62 | 1.53 |
|  | 4 | 1.00 | 0.96 | 0.70 | 9.16 | 7.43 | 6.90 |
|  | 5 | 1.00 | 0.95 | 0.43 | 31.91 | 25.83 | 22.76 |
| Pubmed | 2 | 1.00 | 0.86 | 0.75 | 0.75 | 0.67 | 0.67 |
|  | 3 | 1.00 | 0.91 | 0.47 | 1.24 | 1.11 | 1.05 |
|  | 4 | 1.00 | 0.89 | 0.18 | 2.07 | 1.87 | 1.81 |
|  | 5 | 1.00 | 0.87 | 0.07 | 4.21 | 3.85 | 3.62 |

layers, the baseline models (including the current layerwise sampling methods FastGCN and AS-GCN) can hardly maintain consistently resonable results, and their results drop significantly in deep layers. Their performances are in concordance with our expectations as analyzed in Section 3 which provides the possible explanations. Overall, the comparation results in Table 2 clearly demonstrate the effectiveness of our proposed BLS-GAN model and especially exhibit that our model has the elegant potential to be applied with very deep layers.

Table 3 shows the speedups of all the comparative methods on all the three benchmark datasets by comparing with the training time per epoch of GCN (One training epoch means a complete pass of all training nodes). We can find that by using the idea of layerwise sampling, our proposed BLS-GAN model is computationally efficient with the training time growing linearly with the depth of layers. In detail, on the Reddit dataset, compared with the vanilla models (GCN and GAT) and the nodewise sampling based model (GraphSAGE), our BLS-GAN can achieve about 3x-53x speedup in terms of the training time with different depths of propagation layers, and the deeper the layers, the greater the speedup. Compared with the existing layerwise sampling based models (FastGCN and AS-GCN), although the training time of our model has a slight increase, it is desirable to achieve significant improvements in effectiveness with a slight decrease in efficiency, especially when the model layer is deep.

## 5.3 CASE STUDY

The proposed method BLS-GAN consists of two cores: the bi-diffusion based layer-wise sampling (abbreviated as BLS) and the learnable aggregator based on graph attention mechanism (abbreviated

Table 4: The Micro-F1 scores for node classification by using the bi-diffusion based layer-wise sampling (abbreviated as BLS) and the learnable aggregator based on graph attention mechanism (abbreviated as GAN), respectively.

| Dataset | Reddit | | PPI | | Pubmed | |
|---|---|---|---|---|---|---|
| #Layers | BLS-GCN | AS-GAN | BLS-GCN | AS-GAN | BLS-GCN | AS-GAN |
| 2 | 0.9457 | 0.9384 | 0.8928 | 0.9243 | 0.8864 | 0.8825 |
| 3 | 0.9498 | 0.9434 | 0.9378 | 0.9551 | 0.8905 | 0.8866 |
| 4 | 0.9512 | 0.9448 | 0.9602 | 0.9489 | 0.8926 | 0.8865 |
| 5 | 0.9536 | 0.9402 | 0.9668 | 0.9027 | 0.8942 | 0.8809 |

as GAN). In this section, we conduct extra experiments to disentangle the effects of the two parts of BLS-GAN. We implement two variants of the proposed BLS-GAN: BLS-GCN(using bi-diffusion based sampling, but the constant weights of GCN instead of the attention mechanism) and AS-GAN(using the adaptive sampling of AS-GCN instead of the bi-diffusion based sampling, but the learnable weights of graph attention mechanism). The results are shown in Table 4.

We can see that both the bi-diffusion based sampling and the graph attention mechanism can improve the performance of graph neural networks. For the bi-diffusion based sampling, we can observe that BLS-GCN could achieve significant gains compared to all the baselines on all the datasets, particularly if the layers go deeply. In most cases, the deeper the layers, the greater the gains. For example, compared with the Micro-F1 scores of all the five baselines on the Reddit dataset (as shown in Table 2), BLS-GCN achieves the improvements by about 1%–7% with the setting of 4 layers, and about 3%–8% with the setting of 5 layers. The bi-diffusion based sampling strategy has the elegant potential to be applied with very deep layers. For the graph attention mechanism, we can see that compared with other approaches, AS-GAN can only achieve slight improvements on the Reddit dataset and Pubmed dataset, but can achieve significant gains on the PPI dataset. The results indicate that the attention mechanism is obviously helpful for the embedding learning but the effect may depend on the peculiarity of target dataset. Nonetheless, we can find that the graph attention mechanism is desirable because it can consistently alleviate the decline of results in deep layers on all the three datasets.

## 6 CONCLUSION

In this paper, we first present a novel "bi-diffusion" based layerwise sampling strategy. Distinguishing from existing layerwise sampling algorithms, we sample the nodes layer by layer conditionally based on the factors of the bi-directional diffusion between layers, considering both the influence of parent nodes of upper layer on the candidate nodes of lower layer and the reverse influence of lower candidates on the upper parents. As a result, the sampler tends to construct a mini-batch of nodes with high local bi-directional influence/correlation and thus elegantly and naturally mitigates both the "neighbor explosion" and "over-smoothing" problem, as well as the sparsity issue of current layerwise sampling algorithms. After sampling, we also apply the self-attention mechanism as the aggregator to flexibly learn suitable weights for different-hop nodes during the training, which allows the model to be able to simultaneously exploit both the first-order proximity and the higher-order proximity during a single layer propagation process. Finally, we conduct extensive experiments on three large graphs and the results demonstrate the superiority of our model.

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
