# OpenReview forum: "A bi-diffusion based layer-wise sampling method for deep learning in large graphs"
_ICLR.cc/2020/Conference — Reject_

### Official Review · AnonReviewer2 · 2019-10-23
**Official Blind Review #2**

**Rating:** 6

**Review:**

This paper was an interesting read. The idea of this paper is to challenge the use of Laplacian matrix in GCN. Indeed, typical GCNs use the same adjacency matrix across different layers. In particular, this typically leads in Euclidean case to learning isotropic filters (because the euclidean Laplacian is isotropic). Consequently, such filters have no selectivity at all.(in the Euclidean case, that could correspond to the selectivity to orientations - no selectivity would lead to a difference of Gaussians) Furthermore, for non-sparse graphs, computing the iterations of the Laplacian matrix can require a significant computational power.

In order to tackle this problem, the authors introduced a diffusion factor to sample a set of nodes to build some GCN filters with finite support. At a given layer, the diffusion factors is based on the interaction with other layers of the GCN. Then, a layer-wise attention mechanism that will allow to weight the graph connectivity of the sampled nodes is used, which is supervisedly learned. Each numerical experiments lead to a significantly better accuracy, while the method trains in reasonable time. This is thus numerically convincing. Furthermore, this method is, to my knowledge, new.

The paper is clearly written, the numerical experiments are convincing and the authors address a difficult problem with a simple method: I'm leaning toward an "Accept".

Minor:
- Tables 2/3 are hard to read.
- The paper is 10 pages long, yet this was an interesting read.

Post-discussion:
The other reviewers have made some good point, and thus I decided to lower my score. I still find the paper address an interesting problem.

**Experience Assessment:**

I have read many papers in this area.

**Review Assessment: Checking Correctness Of Derivations And Theory:**

I carefully checked the derivations and theory.

**Review Assessment: Checking Correctness Of Experiments:**

I carefully checked the experiments.

**Review Assessment: Thoroughness In Paper Reading:**

I read the paper thoroughly.

---

> ### Author Response · Authors · 2019-11-11
> **Response to Reviewer #2**
>
> Thank you very much for your comments.
> In the revised manuscript, we adjust the presentation of Table 3 by showing the relative speedups with the results of GCN rather than a long table with numbers, which makes it easier to read. Moreover, we further polish our manuscript, and try our best to make it more concise.
> Thanks again for your comments.

---

### Official Review · AnonReviewer1 · 2019-10-23
**Official Blind Review #1**

**Rating:** 3

**Review:**

The authors propose a sampling method for graph neural networks which is applicable to very large graphs (where not all nodes can be kept in memory at the same time). The method uses the transition probabilities of a random walk to construct a sampling probability of the nodes in the lower layer given the nodes in the upper layer. Since this samples nodes which can be one or multiple hops away, an attention mechanism is used to weight updates from connected nodes. Experiments show that this method is promising.

In its current state I would be inclined to reject this paper, but I could be convinced otherwise. The idea, although relatively straightforward, seems powerful and the experiments seem to support it. My main concern is that paper is not well written. It contains long meandering paragraphs (e.g., all of section 2 is a single paragraph) with high-level intuitions and ill-defined terms, making it hard to read. Similarly, results are badly presented. For example, table 3 should probably be given as relative speedups with the best results bold-faced rather than a long table with numbers. Illustrations would also be very helpful to provide an intuition about formulas 4, 5, and 6. Moreover, a simple ablation study is necessary (e.g., using bi-diffusion based sampling, but using constant weights instead of the attention mechanism, or vice-versa, using the attention mechanism with other types of sampling). It is currently impossible to disentangle the effects of the two parts of BLS-GAN.

**Experience Assessment:**

I have published one or two papers in this area.

**Review Assessment: Checking Correctness Of Derivations And Theory:**

I assessed the sensibility of the derivations and theory.

**Review Assessment: Checking Correctness Of Experiments:**

I assessed the sensibility of the experiments.

**Review Assessment: Thoroughness In Paper Reading:**

I made a quick assessment of this paper.

---

> ### Author Response · Authors · 2019-11-11
> **Official Blind Review #1**
>
> Thank you very much for your insightful and helpful comments on our submitted manuscript. We fully accept these valuable comments and carefully revise the manuscript one by one. The detailed revisions are reported as follows.
>
> 1) We carefully revise the writing of our manuscript by removing the high-level intuitions and ill-defined terms. Moreover, we further polish our manuscript, and try our best to make it easier to read.
>
> 2) We adjust the presentation of Table 3 by showing the relative speedups with the results of GCN rather than a long table with numbers, which makes it more easy to read.
>
> 3) We add a ‘Case Study’ section in the revised manuscript to disentangle the effects of the two parts of BLS-GAN. In the Case Study, we implement two variants of our BLS-GAN: BLS-GCN(using bi-diffusion based sampling, but the constant weights of GCN instead of the attention mechanism) and AS-GAN(using the adaptive sampling of AS-GCN instead of the bi-diffusion based sampling, but the learnable weights of graph attention mechanism). The results demonstrate that: a) the bi-diffusion based sampling could achieve significant gains on all the datasets, and the deeper the layers, the greater the gains; b) the attention mechanism is obviously helpful for the embedding learning but the effect may depend on the peculiarity of target dataset.
>
> Thanks again for your comments, and hope this response could clear your concerns.

---

### Official Review · AnonReviewer4 · 2019-11-08
**Official Blind Review #4**

**Rating:** 3

**Review:**

This paper aims at improving the computational efficiency of GCNs to effectively capture information from the larger multi-hop neighborhood. Conventionally, GCNs use information from all the neighbors up to a certain depth; in which case, with consideration of each further hop, the neighborhood size increases exponentially. To avoid the exponentially increasing memory and computational footprints of GCNs as a result of an exponential neighborhood expansion, this paper proposes a (hop) layer-wise sampling procedure that reduces the complexity to a linear factor. The sampling of nodes at a layer, ‘l’ is based on the transmission probabilities of the nodes at layer ‘l’ and their immediate neighbors sampled earlier in layer ‘l+1’ from both directions of diffusion. The proposed model is based on Graph ATtention Network (GAT) which is adopted here to aggregates neighborhood information only over the nodes sampled with their bi-diffusion sampler.

Strengths of the paper: The paper intuitively suggests that some of the popular sampling-based scaling approaches for GCN may not be powerful enough as they don’t consider bi-directional influences.

Weaknesses of this paper:
- Novelty: The idea is incremental. The paper is similar to the layer-wise sampling model, AS-GCN where instead of the base GCN model this paper uses GAT coupled with its proposed bi-diffusion sampler.
- Experimental results:
    - (a) Inconsistent baseline results: The performance of baselines reported here on standard train/test/val splits are significantly lower than the ones reported in the original papers. For ex: with FastGCN the original papers report 0.88 and 0.937 on PPI and Reddit which is ~0.03 more than what is reported here. With the case of AS-GCN, the original performance scores are superior to the proposed model in the paper, however here they are reported ~0.04 scores lower. Since the codes for all these baselines are available, it is only fair to use the original implementation; if not, it is important to replicate the original results before using a different implementation.
    - (b) Variance and statistical significance results are missing
    - (c) Cluster-GCN though discussed, an experimental comparison with it is missing. Reported results from Cluster-GCN paper on Reddit and PPI suggests a superior performance over BLS-GAT.
    - (d) BLS-GCN missing. This would be a fair comparison to FastGCN and AS-GCN.
    - (e) Experimental comparison with Jumping Neural network (Xu et al) is missing to understand how the proposed solution improves over existing solutions for over-smoothing. It would be helpful to even couple it with Fast-GCN/AS-GCN sampler to better understand the benefits of this paper.
- Writing:
    - The paper is not well written. Though there are only minor grammatical mistakes, multiple sections of the paper are not clear and are hard to read because of complex sentences and long paragraphs.
    - Some of the terminologies used are not clearly described and are not explained prior to the usage. Some of them are neighbor-explosion, over-expansion, the width of neighborhood expansion, local correlations, etc, In some places, over-expansion is used to refer only neighbourhood explosion or only over-smoothing and both. It will be comprehensive if it is grounded.
    - Numerous claims/ideas put forth in this paper are abstract and intuitive. The intuitions should be backed with proper support. Some of the major concerns are:-
        - (a) proof/arguments to show that layer-wise sampling may lead to sparse mini-batches and how does that in-turn impact over-smoothing
        - (b) how does the proposed model avoid over-smoothing?
        - (c) why sub-graph methods are not effective ? .. etc
    - It is true that using a fixed neighborhood weightage function as with GCNs may not be optimal. However, the discussion made on GCN and its lack of an appropriate normalization/ neighbourhood weightage function is incorrect. GCNs aggregate information from further neighborhoods according to the respective higher-order diffusion laplacian matrix entries. You can see that by simply removing the nonlinearity+weights and recursively expanding the GCN equation.
- Other comments:
    - Provide the complexity of the proposed model (GCN + sampler) and compare it with other sampling approaches.
    - The connectivity structure + signal on the nodes of the graphs is the data that is being convolved and they are not the filters. The weights being learned are the filters.
    - In Eqn: 2, I believe you are providing an equation for GS-GCN. In which case the fraction should be N(v)+1/ (N_s(V)+1) to match the original model/implementation for GraphSAGE (GS) paper.
    - I think the summation in the denominator for AS-GCN following Eqn: 3 should run over V instead of V_l.
    - It will be helpful to run the model on directed datasets to see improved benefits of bi-diffusion sampling.
    - Need more discussion about AS-GCN, Cluster-GCN and Jumping Neural networks (Xu 2018b)


**Experience Assessment:**

I have published one or two papers in this area.

**Review Assessment: Checking Correctness Of Derivations And Theory:**

I carefully checked the derivations and theory.

**Review Assessment: Checking Correctness Of Experiments:**

I carefully checked the experiments.

**Review Assessment: Thoroughness In Paper Reading:**

I read the paper thoroughly.

---

> ### Author Response · Authors · 2019-11-11
> **Response to Review #4**
>
> Thank you very much for your insightful and helpful comments on our submitted manuscript. We fully accept these valuable comments and carefully revise the manuscript one by one. The detailed revisions are reported as follows.
>
> 1) We re-implement all the baselines by considering the following three concerns: Firstly, the original implementations of different methods have inconsistent model tricks and data preprocessing, e.g., FastGCN uses the normalization layers after each convolution layer but others not, AS-GCN applies two model designs on the large Reddit dataset and other datasets, respectively, AS-GCN also uses an extra MLP layer for better feature learning. Therefore, it may not be very fair to use the original implementations. Secondly, some original codes (e.g. AS-GCN) have poor flexibility (e.g. adjust the layers). As the convolution layers are hard-coded into the models, it is not easy to extend them from shallow layers (e.g 3 layers) to deep layers (e.g 5 layers). Lastly but most importantly, we develop a general graph deep learning system with flexible extensity, in which we implement all the baselines with a unified paradigm (in fact, most of the existing GNN models can be very easily implemented in our unified framework). We will release the system after review.
>
> 2) We add a ‘Case Study’ section in the revised manuscript to disentangle the effects of the two parts of BLS-GAN. In the Case Study, we implement two variants of our BLS-GAN: BLS-GCN(using bi-diffusion based sampling, but the constant weights of GCN instead of the attention mechanism) and AS-GAN(using the adaptive sampling of AS-GCN instead of the bi-diffusion based sampling, but the learnable weights of graph attention mechanism). The results demonstrate that: a) the bi-diffusion based sampling could achieve significant gains on all the datasets, and the deeper the layers, the greater the gains; b) the attention mechanism is obviously helpful for the embedding learning but the effect may depend on the peculiarity of target dataset.
>
> 3) We adjust the presentation of Table 3 by showing the relative speedups with the results of GCN rather than a long table with numbers, which makes it more easy to read.
>
> 4) We carefully revise the writing of our manuscript by removing the high-level intuitions and ill-defined terms. Moreover, we further polish our manuscript, and try our best to make it easier to read.
>
> Finally, thanks again for your comments, and hope this response could clear your concerns.

---

> > ### Comment · AnonReviewer4 · 2019-11-14
> > **Comments to the author-response**
> >
> > I appreciate the authors for considering the comments and updating the paper and reporting results for some of the additional experiments asked.
> >
> > - On a fair comparison with a GCN base model, BS-GCN, the improvements are not that huge as reported with BS-GAT, they are ~1%. Since the results are not reported over multiple runs (different seeds, ideally different train/test sets) the significance of this improvement is not clear.
> > - Moreover, I still stand with my statement that it is not fair to report use a setting for baselines where their performance is lower than reported except for the following two cases: 1) the case where the original implementation of the baselines yield a lower performance similar to what the authors report here, i.e if the baseline results are not reproducible 2) the case where the authors convince that the baseline setting is not acceptable for certain reasons . Otherwise,  I would suggest that the authors use the settings of the baseline models to be fair for the proposed model. Unless one of the following condition is the case,

---

### Decision · Program_Chairs · 2019-12-19

**Decision:**

Reject

**Comment:**

This paper addresses the challenge of time complexity in aggregating neighbourhood information in GCNs. As we aggregate information from larger hops (deeper neighbourhoods) the number of nodes can increases exponentially thereby increasing time complexity. To overcome this the authors propose a sampling method which samples nodes layer by layer based on bidirectional diffusion between layers. They demonstrate the effectiveness of their approach on 3 large benchmarks.

While the ideas presented in the paper were interesting the reviewers raised some concerns which I have summarised fellow:

1) Novelty: The reviewers felt that the techniques presented were not very novel and is very similar to one existing work as pointed out by R4
2) Writing: The writing needs to be improved. The authors have already made an attempt towards this but it could be improved further
3) Comparisons with baselines: R4 has raised some concerns  the settings/configurations used for the baseline methods. In particular, the results for the baseline methods are lower than those reported in the original papers. I have read the author's rebuttal for this but I am not completely convinced about it. I would suggest that the authors address this issue in subsequent submissions

Based on the above reasons I recommend that the paper cannot be accepted.